# A caspase–RhoGEF axis contributes to the cell size threshold for apoptotic death in developing *Caenorhabditis elegans*

**Aditya Sethi**[1,2☯], **Hai Wei**[1☯¤a], **Nikhil Mishra**[1¤b], **Ioannis Segos**[2], **Eric J. Lambie**[2], **Esther Zanin**[1,3], **Barbara Conradt**[2]*

**1** Faculty of Biology, Center for Integrative Protein Sciences Munich (CIPSM), Ludwig-Maximilians-University Munich, Planegg-Martinsried, Germany, **2** Department of Cell & Developmental Biology, Division of Biosciences, University College London, London, United Kingdom, **3** Department Biology, Friedrich-Alexander-University Erlangen-Nürnberg, Erlangen, Germany

☯ These authors contributed equally to this work.
¤a Current address: University of Texas Southwestern Medical Center, Dallas, Texas, United States of America
¤b Current address: Institute of Science and Technology Austria, Klosterneuburg, Austria
* b.conradt@ucl.ac.uk

**Data Availability Statement:** All relevant data are within the paper and its Supporting information files.

## Abstract

A cell's size affects the likelihood that it will die. But how is cell size controlled in this context and how does cell size impact commitment to the cell death fate? We present evidence that the caspase CED-3 interacts with the RhoGEF ECT-2 in *Caenorhabditis elegans* neuroblasts that generate "unwanted" cells. We propose that this interaction promotes polar acto-myosin contractility, which leads to unequal neuroblast division and the generation of a daughter cell that is below the critical "lethal" size threshold. Furthermore, we find that hyperactivation of ECT-2 RhoGEF reduces the sizes of unwanted cells. Importantly, this suppresses the "cell death abnormal" phenotype caused by the partial loss of *ced-3* caspase and therefore increases the likelihood that unwanted cells die. A putative null mutation of *ced-3* caspase, however, is not suppressed, which indicates that cell size affects CED-3 caspase activation and/or activity. Therefore, we have uncovered novel sequential and reciprocal interactions between the apoptosis pathway and cell size that impact a cell's commitment to the cell death fate.

## Introduction

Apoptosis is a type of programmed cell death that is conserved throughout the animal kingdom. The pathway that triggers apoptosis in "unwanted" cells includes pro- and anti-apoptotic members of the Bcl-2 family of proteins (BH3-only and Bcl-2-like proteins, respectively), Apaf-1-like adaptor proteins (which form the apoptosome) and members of the caspase family of cysteine proteases [1,2]. In unwanted cells, BH3-only proteins become active, and this leads to apoptosome assembly and the activation of caspases. Once caspase activity has reached a critical "lethal" threshold, apoptosis is triggered. A hallmark of cells undergoing apoptosis is a decrease in cell size [3]. This decrease is likely induced by the opening of potassium and

**Funding:** Some strains were provided by the Caenorhabditis Genetics Center (CGC), which is funded by NIH Office of Research Infrastructure Programs (https://orip.nih.gov/) (P40 OD010440). This work was supported by UCL (Capital Equipment Fund, CEF2), a predoctoral fellowship from the China Scholarship Council (https://www.csc.edu.cn/) to HW, a predoctoral fellowship from the Studienstiftung des Deutschen Volkes (https://www.studienstiftung.de/) to NM, a Wolfson Fellowship from the Royal Society (https://royalsociety.org/) to BC (RSWF\R1\180008), the Deutsche Forschungsgemeinschaft (https://www.dfg.de/en/index.jsp) (ZA619/3-1 and ZA619/3-2 to EZ; C0204/10-1 and EXC114 to BC), and the Biotechnology and Biological Sciences Research Council (https://bbsrc.ukri.org/) (BB/V007572/1 to BC). The funders had no role in study design, data collection and analysis, decision to publish, or preparation of the manuscript.

**Competing interests:** The authors have declared that no competing interests exist.

**Abbreviations:** DH, Dbl-Homology; GEF, guanine nucleotide exchange factor; gf, gain-of-function; lf, loss-of-function; NSM, neurosecretory motor neuron; PH, Pleckstrin Homology; PSP, photostimulable phosphor; ROI, region of interest; ts, temperature-sensitive.

chloride channels in the plasma membrane, causing an efflux of potassium and chloride ions followed by water [4]. Blocking this shrinkage in apoptotic cells in *Caenorhabditis elegans* embryos compromises their ability to die, which suggests that cell shrinkage facilitates the execution of apoptosis [5].

Interestingly, within populations of animal cells grown in culture, smaller cells have a higher likelihood to undergo apoptosis [6]. Furthermore, decreasing the sizes of tissue culture cells by treating them with hypertonic solutions can cause these cells to die through apoptosis [4]. In addition, HeLa cells sometimes divide unequally and generate a smaller and a larger daughter cell. A certain proportion of the smaller daughter cells subsequently undergoes apoptosis [7]. These observations suggest that at least in vitro, a decrease in cell size can also trigger the apoptotic death of a cell. The following observations support the view that cell size can trigger apoptosis also in vivo. In some mutant backgrounds, *Drosophila melanogaster* germline stem cells divide unequally and generate a smaller and a larger daughter cell. As in HeLa cells, a certain proportion of the smaller daughter cells subsequently undergoes apoptosis [8]. And, many of the unwanted cells that reproducibly die through apoptosis during *C. elegans* development are the smaller daughter of a blast cell that divides unequally by size [9,10]. Importantly, mutations that cause such a "mother" to divide equally, thereby causing an increase in the size of the smaller daughter cell, compromise the ability of the unwanted cell to die [11,12]. To give an example, approximately 410 min after the first cleavage of the *C. elegans* 1-cell embryo, the neurosecretory motor neuron (NSM) neuroblast (NSM neuroblast; referred to as "NSMnb") divides unequally. Its larger daughter cell survives and differentiates into the NSM neuron, whereas its smaller daughter cell, the "NSM sister cell" (NSMsc), dies. The loss of the *pig-1* (*pig*, *par-1* like-gene) gene, which encodes a kinase similar to mammalian MELK (maternal embryonic leucine zipper kinase), causes the NSMnb to divide equally resulting in daughter cells of almost identical sizes [11,13]. As a result, the now larger NSMsc sometimes fails to die. Hence, across animal species, there appears to be a critical "lethal size" threshold. Below this threshold, apoptosis can be triggered in cells that normally live. Conversely, above this threshold, apoptosis can be blocked in cells that are programmed to die. How cell size is controlled in this context and how cell size impacts a cell's commitment to the apoptotic fate remains unclear.

We are studying the unequal division of mothers of cells "programmed to die" in *C. elegans*, including the unequal division of the NSMnb. Previously, we proposed that unequal NSMnb division is the result of transient polar cortical contractility of the actomyosin network in the NSMnb prior to its division and local extension of the plasma membrane during NSMnb division [14]. Furthermore, we obtained evidence that the apoptosis pathway is active at a low, nonlethal level in mothers of cells programmed to die [15–17]. Surprisingly, we also found that loss-of-function (lf) mutations of the BH3-only gene *egl-1* (*egl*, egg-laying defective), the Apaf-1-like gene *ced-4* (*ced*, cell-death abnormal), or the caspase gene *ced-3* compromise the ability of mothers to divide unequally and to generate a daughter cell with a size below the critical lethal threshold [16]. To elucidate the mechanism(s) through which the *C. elegans* apoptosis pathway affects the unequal division of mothers, we used a deep-sequencing coupled yeast 2 hybrid screen to search for physical interactors of its downstream effector, the caspase CED-3. In this screen, we identified ECT-2, a guanine nucleotide exchange factor (GEF) of RhoA-type GTPases [18–20]. Through molecular and genetic studies of the interactions between CED-3 caspase and ECT-2 RhoGEF, we have uncovered a novel role of *ced-3* caspase in the control of the actomyosin network in the context of unequal cell division. Our findings also provide in vivo evidence for the existence of an inverse correlation between a cell's size and its likelihood to undergo apoptosis and suggest that cell size affects the activation and/or the activity of CED-3 caspase.

## Results

### CED-3 caspase physically interacts with ECT-2 RhoGEF

To identify physical interactors of the caspase CED-3 that might link its function to unequal cell division, we performed a yeast 2 hybrid screen using a next-generation sequencing based screening procedure (https://nextinteractions.com/). As a bait construct, we used the full-length *ced-3* open reading frame harboring a missense mutation that results in the production of proCED-3 zymogen lacking the critical active site cysteine (proCED-3(C358S)); this protein will be unable to mature into the fully active enzyme (Fig 1A). The proCED-3(C358S) bait was tested for interactions with proteins produced from a *C. elegans* cDNA library that was generated from mRNAs isolated from 8 different developmental stages and that represents approximately 13,000 genes. Among the proteins found to interact with proCED-3(C358S) is ECT-2 (ECT, epithelial cell transforming 2), the *C. elegans* orthologue of the mammalian proto-oncoprotein Ect2, a GEF of the RhoA family of small GTPases [18–20]. The proCED-3(C358S)–ECT-2 interaction is at least 16-fold stronger than the interaction of proCED-3(C358S) with either of 2 negative controls (empty bait-vector control and cDNA control).

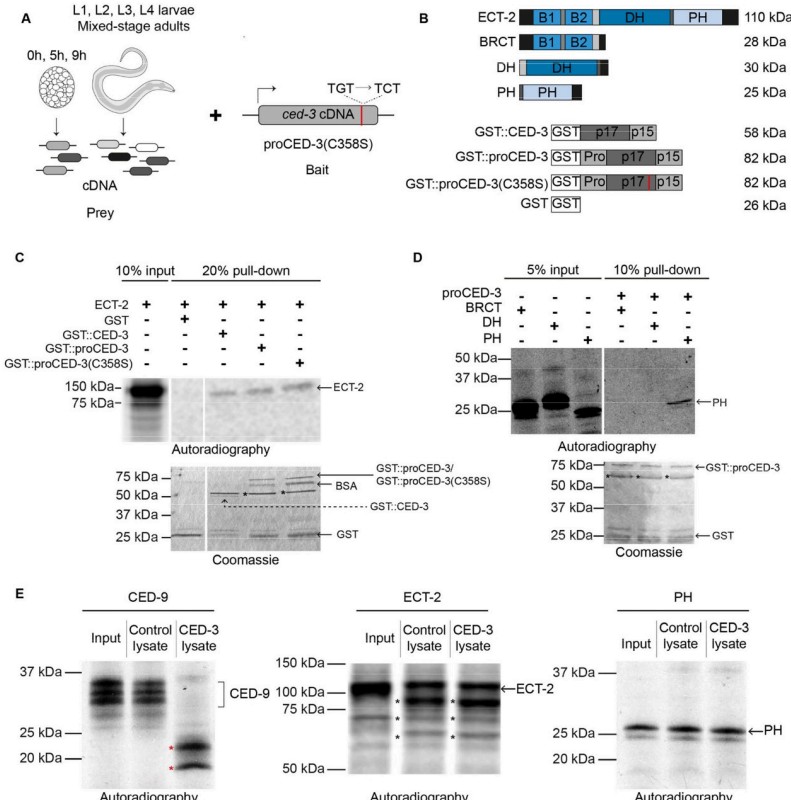

**Fig 1. CED-3**caspase **physically interacts with ECT-2**RhoGEF **in vitro. (A)** Schematic representation of the yeast 2-hybrid screen performed by Next Interactions (https://nextinteractions.com/) to identify physical interactors of proCED-3(C358S). **(B)** Schematic representation of the various constructs used in the GST pull-down assay along with their expected molecular weight in kilodalton (kDa) on the left. **(C and D)** Autoradiographs and Coomassie-stained gels of representative GST pull-down experiments. Black asterisks indicate potential breakdown products resulting from growing recombinant proteins in bacterial cultures. The different lanes shown in the figure are from the same gel. **(E)** Autoradiographs of representative in vitro cleavage experiments with CED-9, ECT-2, and ECT-2's PH domain. Red asterisks indicate CED-9 cleavage products. Black asterisks indicate potential ECT-2 cleavage products resulting from incubation with bacterial lysate.

To confirm that proCED-3(C358S) interacts with ECT-2, we produced GST-tagged proCED-3(C358S) in *E. coli* (GST::proCED-3(C358S)), purified it and incubated it with in vitro translated, [35]S-methinonine-labeled and tagged (S·TAG) full-length ECT-2 protein ([35]S-S·TAG::ECT-2). We found that [35]S-S·TAG::ECT-2 co-purifies with GST::proCED-3 (C358S) but not GST alone (Fig 1B and 1C). Furthermore, we found that [35]S-S·TAG::ECT-2 also co-purifies with recombinant GST::proCED-3 and GST::CED-3, each of which presumably can mature into the fully active CED-3 caspase (Fig 1B and 1C).

ECT-2 has an N-terminal domain with 2 "BRCA1 C Terminus" (BRCT) motifs, a C-terminal "Pleckstrin Homology" (PH) domain and a central "Dbl-Homology" (DH) domain, which includes the catalytic GEF (http://www.uniprot.org/uniprot/Q9U364) [21] (Fig 1B). To determine which of these ECT-2 domains interact(s) with CED-3, we tested each domain for binding to GST::proCED-3. We found that only the C-terminal PH domain co-purifies with GST:: proCED-3 (Fig 1D). In summary, our results demonstrate that both proCED-3 zymogen and active CED-3 caspase can physically interact with ECT-2 RhoGEF in yeast and in vitro, and that these interactions are mediated by ECT-2's PH domain. Therefore, proCED-3 and CED-3 caspase may interact with ECT-2 RhoGEF in vivo.

ECT-2 RhoGEF has 6 predicted caspase cleavage sites, including a cleavage site in the PH domain that is conserved across *Caenorhabditis* species (S1 Fig). To determine whether ECT-2 RhoGEF is a proteolytic substrate of CED-3 caspase, we used an in vitro cleavage assay based on bacterially expressed FLAG-tagged CED-3 protein (CED-3::8xFLAG). As a positive control, we used in vitro translated, [35]S-methionine-labeled and tagged (S·TAG) CED-9 protein ([35]S-S·TAG::CED-9), which was previously shown to be a proteolytic substrate of CED-3 [22]. We found that [35]S-S·TAG::CED-9 is efficiently cleaved by CED-3 (Fig 1E). In contrast, in vitro translated, [35]S-methinonine-labeled full-length ECT-2 ([35]S-S·TAG::ECT-2) or the ECT-2 PH domain ([35]S-S·TAG::PH) are not cleaved by CED-3 using this in vitro assay. These results do not support the idea that an interaction between CED-3 caspase and ECT-2 RhoGEF in vivo results in CED-3 caspase-dependent cleavage of ECT-2 RhoGEF.

## *ect-2* RhoGEF cooperates with *ced-3* caspase to control the size of the NSM sister cell

The *ect-2* RhoGEF gene has been shown to play an important role in the division of the *C. elegans* 1-cell embryo, which like the division of mothers of cells programmed to die is unequal by size and generates 2 daughter cells with different fates. Specifically, *ect-2* RhoGEF promotes polarization of the cortical actomyosin network, which is required for the establishment and maintenance of anterior-posterior PAR protein asymmetry prior to the 1-cell embryo's first division [23–25]. *ect-2* also plays a critical role in cytokinesis and, hence, is an essential gene [20,26]. To determine whether *ect-2* RhoGEF activity also impacts the divisions of mothers of cells programmed to die, we analyzed the effects of a temperature-sensitive (ts) partial lf mutation of *ect-2*, *ax751* [27], on the division of the NSM neuroblast (NSMnb). (The *ax751* mutation causes a single amino acid change C-terminal to the PH domain (G738R) [S2A Fig].) The NSMnb was identified in comma to 1 ½ fold stage embryos based on position and cell shape using a transgene that labels cell boundaries ($P_{pie-1}mCherry::PH^{PLC\Delta}$ [*ltIs44*]) [28] (Fig 2A). We followed the division of individual NSMnb, estimated the sizes of the NSM and NSM sister cell (NSMsc) immediately post-cytokinesis, and divided the size of the NSMsc by the size of the NSM to acquire the "daughter cell size ratio" (Fig 2A). (There are 2 NSMnb, the left and the right NSMnb. Since these 2 neuroblasts are functionally identical, for simplicity, we will refer to them as "NSMnb.") In wild-type animals, we found that daughter cell size ratios range from 0.61 to 0.70 with a mean ratio of 0.66; i.e., the NSMsc is approximately 0.66 times the size of

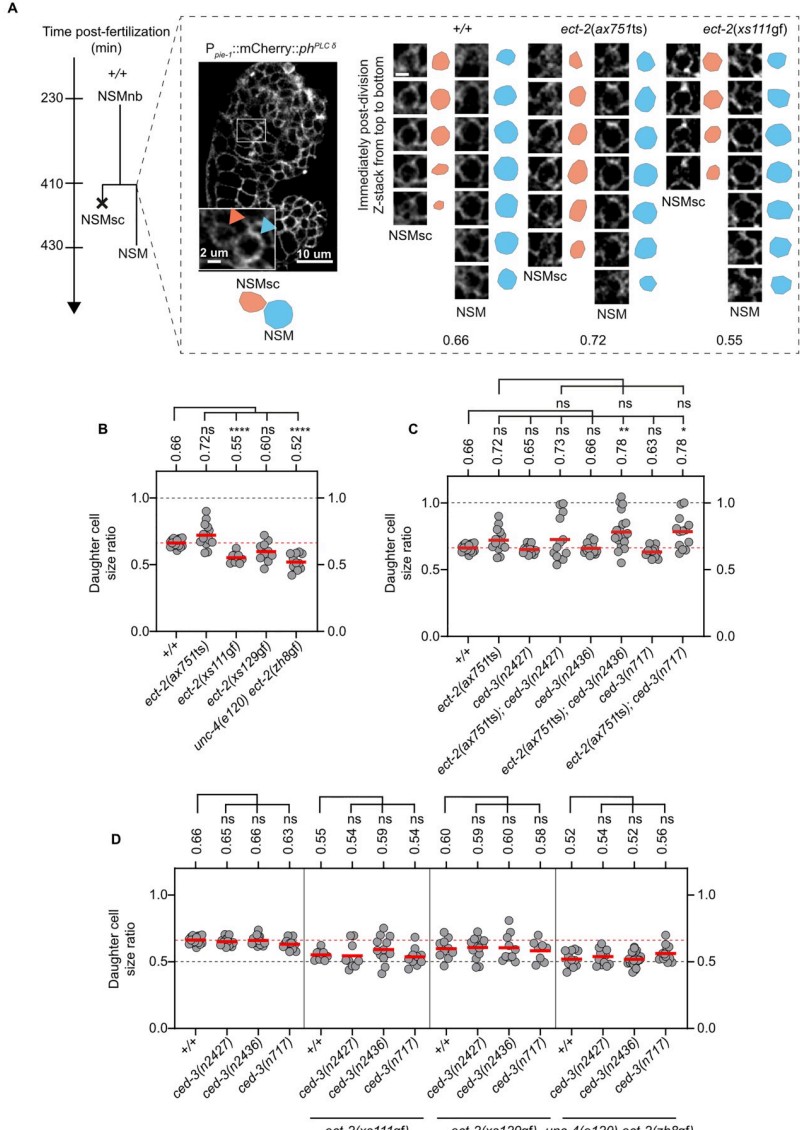

**Fig 2. *ced-3* caspase cooperates with *ect-2* RhoGEF in the control of daughter cell sizes in the NSM lineage. (A)** Schematic representation of the NSM lineage. The NSMsc and NSM can be identified in comma stage embryos using the transgene $P_{pie-1}$::*mCherry::PH^{PLCΔ}* (*ltIs44*), which labels the plasma membrane of cells (orange arrow indicates the NSMsc and blue arrow indicates the NSM). Using confocal imaging, a Z-stack of the NSMsc and NSM can be obtained immediately post-division and the size ratio of the NSMsc:NSM can be estimated. The Z-stack of a pair of NSMsc (orange) and NSM (blue) in +/+, *ect-2(ax751*ts*)* and *ect-2(xs111*gf*)* mutants is shown. The corresponding mean daughter cell size ratios (NSMsc:NSM) are given below. Scale bars: 10 μm and 2 μm. **(B–D)** Daughter cell size ratios in +/+ and various *ect-2* and *ced-3* single and double mutants measured using *ltIs44* (*n* = 10–20). Each gray dot represents the daughter cell size ratio of 1 pair of daughter cells. Horizontal red lines represent mean values, which are also indicated on top. The horizontal red dotted line represents the mean daughter cell size ratio of wild-type (+/+) embryos for comparison. The horizontal black dotted line in Fig 2B and 2C represents a daughter cell size ratio of 1.0 indicating equal division. The horizontal black dotted line in Fig 2D represents a daughter cell size ratio of 0.5 indicating that the smaller daughter is twice as small as the larger daughter. Statistical significance was determined using the Dunnett's T3 multiple comparisons test (**** = $P < 0.0001$, ** = $P < 0.01$, * = $P < 0.05$, ns = $P > 0.05$). NSM, neurosecretory motor neuron.

the NSM (Fig 2B). We found that in *ect-2*(*ax751*ts) animals that were shifted to the nonpermissive temperature (25°C) approximately 2 hours prior to the division of the NSMnb, the mean ratio is 0.72 (range 0.59 to 0.90). Next, we tested 3 different *ced-3* lf mutations for interactions with *ect-2*(*ax751*ts): the weak lf mutation *n2427*, the intermediate lf mutation *n2436*, and the putative null mutation *n717* (S2B Fig) [29,30]. We found that all 3 mutations increase the range of ratios observed in *ect-2*(*ax751*ts) animals (0.62 to 1.00 in the case of *n2427*, 0.55 to 1.05 in the case of *n2426*, and 0.54 to 1.00 in the case of *n717*) (Fig 2C). Importantly, in each of the double mutants, we observed cases in which the NSMnb divided equally, generating 2 daughter cells of essentially identical sizes. In addition, *ced-3*(*n2436*) and *ced-3*(*n717*) both increase the mean ratio of *ect-2*(*ax751*ts) from 0.72 to 0.78, which is significantly different from the mean ratio of 0.66 observed in wild type. These results demonstrate that decreasing *ect-2* RhoGEF function impacts the unequal division of the NSMnb, resulting in larger NSMsc. Furthermore, reduction of *ced-3* caspase function enhances the *ect-2* lf phenotype. Based on these findings, we conclude that *ced-3* caspase and *ect-2* RhoGEF cooperate during unequal NSMnb division to ensure that the size of the NSMsc is below the critical lethal threshold.

## *ect-2* RhoGEF acts downstream of or in parallel to *ced-3* caspase to control the size of the NSM sister cell

To test whether increasing *ect-2* RhoGEF function also impacts unequal NSMnb division, we took advantage of 3 gain-of-function (gf) mutations of *ect-2*, *xs111*gf, *xs129*gf, and *zh8*gf, which cause single amino acid changes in residues (conserved in *Caenorhabditis* species) in the first BRCT motif (E129K), the second BRCT motif (E225K), or the PH domain (G707D), respectively (S2A Fig) [19,31]. We found that all 3 mutations affect the range of daughter cell size ratios. For example, in *ect-2*(*zh8*gf) animals, the ratios range from 0.42 to 0.60, which indicates that some divisions generated an NSMsc that is less than half the volume of the NSM (Fig 2B). (Note, *unc-4*(*e120*) has no effect on daughter cell size ratio [S3 Fig].) Furthermore, *ect-2* (*xs111*gf) and *ect-2*(*zh8*gf) significantly decrease the mean daughter cell size ratio from 0.66 to 0.52 and 0.55, respectively. We also tested the 3 *ced-3* lf mutations for interactions with the *ect-2* gf mutations and found that overall, reducing *ced-3* function has no significant effect on daughter cell size ratios in *ect-2* gf mutants (Fig 2D). These results demonstrate that increasing *ect-2* RhoGEF function impacts unequal NSMnb division, resulting in smaller NSMsc. Reduction in *ced-3* caspase function, however, has no effect on the *ect-2* gf phenotype. This suggests that *ect-2* RhoGEF acts in parallel to, or downstream of, *ced-3* caspase to ensure that the size of the NSMsc is below the critical lethal threshold.

## Active CED-3 caspase is required for polar localization of ECT-2 RhoGEF protein in the NSM neuroblast prior to its unequal division

We previously proposed that polar cortical contractility mediated by the actomyosin network is required for unequal NSMnb division. Specifically, we showed that starting approximately 5 minutes before metaphase–anaphase transition, the nonmuscle myosin II NMY-2 is found cortically enriched on the ventral side of the NSMnb. This is the side of the NSMnb that subsequently forms the NSM, the larger daughter cell, which survives [14]. Furthermore, we showed that reducing the function of the *nmy-2* gene (using a partial ts lf mutation of *nmy-2* nonmuscle myosin II) or abolishing the ventral cortical enrichment of the NMY-2 protein in the NSMnb (using an lf mutation of *pig-1* MELK) compromises unequal NSMnb division and results in the generation of 2 daughter cells of essentially identical sizes [14]. Prior to the first division of the *C. elegans* 1-cell embryo, ECT-2 RhoGEF is found enriched on the plasma membrane on the anterior side (which is the side that subsequently forms AB, the larger

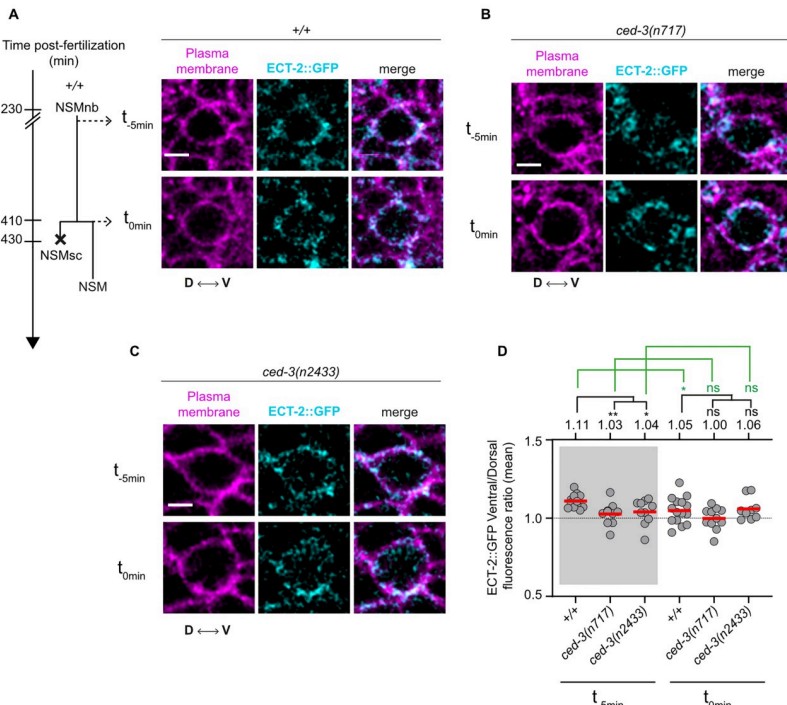

**Fig 3. *ced-3* caspase is required for the asymmetric enrichment of ECT-2 RhoGEF in the NSM neuroblast. (A)** Schematic representation of the NSM lineage indicating the 2 time points ($t_{-5min}$ = 5 minutes before metaphase, $t_{0min}$ = metaphase) of the NSM neuroblast used for imaging and central Z-slices of representative wild-type (+/+) NSM neuroblasts expressing the transgene *ltIs44* ($P_{pie-1}$::*mCherry*::$PH^{PLC\Delta}$) (magenta) and CRISPR allele *zhIs135* (*ect-2*::*zf-1*::*gfp*) (cyan). D indicates the dorsal side and V the ventral side of the NSMnb. Scale bar: 2 μm. **(B and C)** Representative images of central Z-slices of representative *ced-3(n717)* and *ced-3(n2433)* NSM neuroblasts expressing the transgene *ltIs44* ($P_{pie-1}$::*mCherry*::$PH^{PLC\Delta}$) (magenta) and CRISPR allele *zhIs135* (*ect-2*::*zf-1*::*gfp*) (cyan) at 2 time points ($t_{-5min}$ = 5 minutes before metaphase, $t_{0min}$ = metaphase). Scale bar: 2 μm. **(D)** Ventral/dorsal ratios of mean GFP fluorescence intensities in the NSM neuroblast in animals of indicated genotypes carrying the CRISPR allele *zhIs135* (*ect-2*::*zf-1*::*gfp*). Each gray dot represents the ventral/dorsal fluorescence intensity ratio of 1 NSM neuroblast ($n = 10–15$). The mean values are indicated by the horizontal red lines and are also given on top. The horizontal black dotted line represents a fluorescence intensity ratio of 1, which indicates no asymmetry in fluorescence intensity between the ventral and dorsal side of the NSM neuroblast. Statistical significance is indicated on top. The black lines represent statistical significance comparing the wild-type to *ced-3(n717)* and *ced-3(n2433)*. The green dotted lines represent statistical significance comparing the 2 time points ($t_{-5min}$ = 5 minutes before metaphase, $t_{0min}$ = metaphase) of the same genotype. Statistical significance was determined using the Welch's 2 sample *t* test (** = $P < 0.01$, * = $P < 0.05$, ns = $P > 0.05$). NSM, neurosecretory motor neuron.

daughter cell) where it contributes to actomyosin-dependent cortical contractility [23,24]. For this reason, we analyzed ECT-2 RhoGEF localization in the NSMnb using ECT-2 protein endogenously tagged at the C terminus with GFP (ECT-2::GFP) (*ect-2(zh135)*) [32] (S4 Fig). We found that approximately 5 minutes prior to metaphase–anaphase transition ($t_{-5min}$), ECT-2::GFP is significantly enriched on the ventral side of the NSMnb (Fig 3A and 3D and S4 Fig). This enrichment is not observed approximately 10 minutes prior to metaphase–anaphase transition ($t_{-10min}$), and it is no longer observed at metaphase–anaphase transition ($t_{0min}$) (Fig 3A and S5 Fig). As a control, we measured the signal of the cell boundary marker $P_{pie-1}$*mCherry*::$PH^{PLC\Delta}$ in the ventral and dorsal side of the NSMnb and did not find an enrichment on the ventral side of the NSMnb at either $t_{-5min}$ or $t_{0min}$ (S6 Fig). Importantly, the putative *ced-3* null mutation *n717* results in the loss of the ventral enrichment of ECT-2::GFP at $t_{-5min}$ (Fig 3B and 3D). In contrast, *ced-3(n717)* has no effect on the ventral cortical enrichment in the NSMnb of NMY-2::GFP (Fig 4A and 4C). Similarly, *ced-3(n717)* has no effect on the

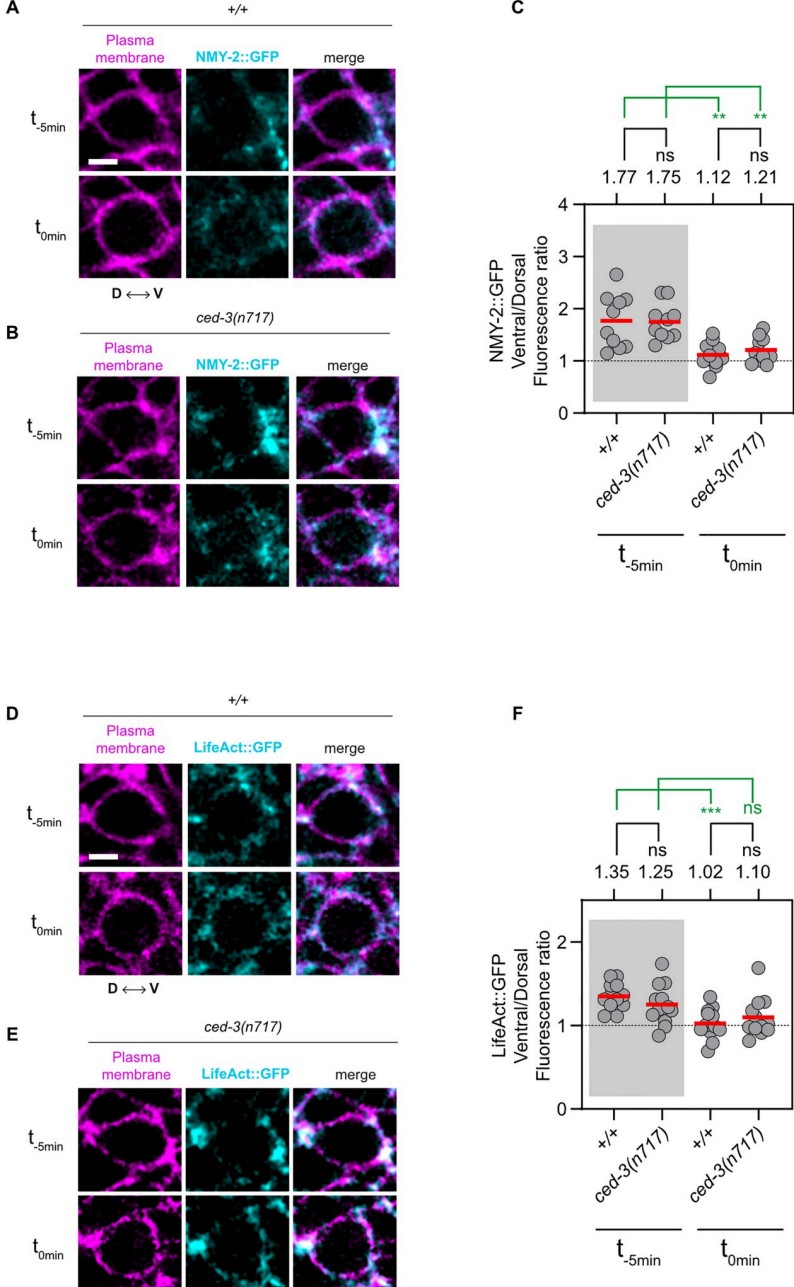

**Fig 4. Asymmetric enrichment of NMY-2 and F-actin in the NSM neuroblast is not dependent on *ced-3* caspase.**
**(A and B)** Representative images of central Z-slices of representative wild-type (+/+) and *ced-3(n717)* NSM
neuroblasts expressing the transgene *ltIs44* ($P_{pie-1}$::*mCherry*::$PH^{PLCΔ}$) (magenta) and CRISPR allele *cp13* (*nmy-2::gfp*
+ *LoxP*) (cyan) at 2 time points ($t_{-5min}$ = 5 minutes before metaphase, $t_{0min}$ = metaphase). D is the dorsal side and V is
the ventral side. Scale bar: 2 μm. **(C and F)** Ventral/dorsal ratios of mean GFP fluorescence intensities in the NSM
neuroblast in animals of indicated genotypes carrying the CRISPR allele *cp13* (*nmy-2::gfp* + *LoxP*) (C) or the transgene
*ddIs86* ($P_{pie-1}$::*LifeAct::gfp*) (F). Each gray dot represents the ventral/dorsal fluorescence intensity ratio of 1 NSM
neuroblast (*n* = 10–15). The mean values are indicated by the horizontal red lines and are also given on top. The
horizontal black dotted line represents a fluorescence intensity ratio of 1, which indicates no asymmetry in
fluorescence intensity between the ventral and dorsal side of the NSM neuroblast. Statistical significance is indicated
on top. The black lines represent statistical significance comparing the wild-type to *ced-3(n717)* and *ced-3(n2433)*. The
green lines represent statistical significance comparing the 2 time points ($t_{-5min}$ = 5 minutes before metaphase, $t_{0min}$ =
metaphase) of the same genotype. Statistical significance was determined using the Welch's 2 sample *t* test (*** =
$P < 0.001$, ** = $P < 0.01$, ns = $P > 0.05$). **(D and E)** Representative images of central Z-slices of representative wild-

type (+/+) and *ced-3(n717)* NSM neuroblasts expressing the transgenes *ltIs44* ($P_{pie-1}$::*mCherry*::$PH^{PLCΔ}$) (magenta) and *ddIs86* ($P_{pie-1}$::*LifeAct*::*gfp*) (cyan) at 2 time points (t$_{-5min}$ = 5 minutes before metaphase, t$_{0min}$ = metaphase). D is the dorsal side and V is the ventral side. Scale bar: 2 μm. NSM, neurosecretory motor neuron.

localization in the NSMnb of F-actin (visualized using LifeAct::GFP), which we found to be cortically enriched on the ventral side similar to NMY-2 nonmuscle myosin II (Fig 4B and 4D). As shown above (Fig 1), ECT-2 RhoGEF can physically interact with both the proCED-3 zymogen and the matured, active CED-3 caspase in vitro. To determine whether the enrichment of ECT-2::GFP on the ventral side of the NSMnb is dependent on CED-3 caspase activity, we analyzed animals homozygous for the *ced-3* missense mutation, *n2433*. This mutation causes an amino acid change (G360S) that disrupts the active site of CED-3 caspase. proCED-3 (G360S) zymogen is thus expected to be unable to mature into the fully active enzyme (S2B Fig) [30]. We found that like *ced-3(n717)*, *ced-3(n2433)* abolishes the ventral enrichment of ECT-2::GFP in the NSMnb prior to its division (Fig 3C and 3D). In summary, these results show that approximately 5 minutes prior to metaphase–anaphase transition, there is a transient enrichment of ECT-2::GFP on the ventral side of the NSMnb and that this transient enrichment requires active CED-3 caspase. This finding provides support for the notion that in the context of the unequal NSMnb division, *ect-2* RhoGEF acts downstream of, rather than in parallel to, *ced-3* caspase. Furthermore, it suggests that *ced-3* caspase and *ect-2* RhoGEF impact unequal NSMnb division by promoting actomyosin-dependent cortical contractility on the ventral side of the NSMnb.

## The *ced-3* caspase, *ect-2* RhoGEF-dependent pathway acts in parallel to the *pig-1* MELK, *nmy-2* nonmuscle myosin II-dependent pathway to control the size of the NSM sister cell

The cortical enrichment of nonmuscle myosin II NMY-2 protein approximately 5 minutes prior to NSM neuroblast division is dependent on *pig-1* MELK [14] but not *ced-3* caspase (Fig 4A). In addition, we previously found that the loss of *ced-3* caspase in animals homozygous for the strong *pig-1* MELK lf mutation, *gm344* [11], increases the mean daughter cell size ratio in the NSM lineage from 1.0 to 1.25, which indicates that the NSMsc is now larger than the NSM [16]. In contrast, the loss of the gene *strd-1* STRADα, which acts in a *par-4* LKB-dependent pathway required for the activation of PIG-1 MELK kinase activity [32–34], does not increase the daughter cell size ratio in the NSM lineage in a *pig-1(gm344)* background (1.01 compared to 1.02; [14]). These observations suggest that *ced-3* caspase acts in parallel to the *pig-1* MELK, *nmy-2* nonmuscle myosin II-dependent pathway to ensure that the size of the NSMsc is below the critical lethal threshold. To determine whether reducing *ect-2* RhoGEF function also increases the daughter cell size ratio in *pig-1(gm344)* animals, we attempted to generate animals homozygous for both *ect-2(ax751*ts) and *pig-1(gm344)*. Unfortunately, we were unable to obtain such a strain, suggesting that animals lacking both genes are not viable. However, we found that animals homozygous for *ect-2(ax751*ts) and *ok2283*, a strong lf mutation of *strd-1* STRADα [35], are viable. To determine whether reducing *ect-2* RhoGEF function increases the daughter cell size ratio in animals defective in the *pig-1* MELK, *nmy-2* nonmuscle myosin II-dependent pathway, we therefore analyzed *ect-2(ax751*ts); *strd-1(ok2283)* animals. We found that *ect-2(ax751*ts) increases the mean daughter cell size ratio in *strd-1(ok2283)* animals from 0.95 to 1.11, which is statistically significant (S7 Fig). Furthermore, we found that the *ect-2* gf mutation *xs111* reduces the mean daughter cell size ratio in *strd-1(ok2283)* animals from 0.95 to 0.83 (S7 Fig). (Of note, we were also unable to generate the strain *ect-2(xs111*gf); *pig-1 (gm344)*, suggesting that it is also not viable.) Based on these observations, we propose that the

*ced-3* caspase, *ect-2* RhoGEF-dependent pathway acts in parallel to the *pig-1* MELK, *nmy-2* nonmuscle myosin II-dependent pathway to control the size of the NSMsc and to ensure that its size is below the critical lethal threshold.

## *ect-2* RhoGEF has pro-apoptotic activity, and this pro-apoptotic activity is dependent on *ced-3* caspase

Decreasing or increasing *ect-2* RhoGEF function affects the unequal division of the NSMnb, resulting in NSMsc that are larger or smaller in size, respectively. To determine whether decreasing or increasing *ect-2* RhoGEF function affects the likelihood of the NSMsc to die, we analyzed larvae of the fourth larval stage (L4 larvae) carrying a NSM reporter, P$_{tph-1}$*his-24::gfp*. This reporter is expressed in NSM neurons and also in inappropriately surviving "undead" NSMsc (Fig 5A) [36,37]. In wild-type animals, the NSMsc always dies, resulting in 0% NSMsc survival (Fig 5B). In contrast, in *ced-3*(*n717*) animals, NSMsc survival is nearly 100%. Using *ect-2*(*ax751*ts), we found that the partial loss of *ect-2* RhoGEF causes 4% NSMsc survival. In addition, we found that *ect-2*(*ax751*ts) increases NSMsc survival from 18% to 25% in animals homozygous for the weak *ced-3* lf mutation *n2427* and from 68% to 81% in animals homozygous for the intermediate *ced-3* lf mutation *n2436* (Fig 5B). Animals homozygous for the 3 *ect-2* RhoGEF gf mutations do not exhibit NSMsc survival in a wild-type background (0%; Fig

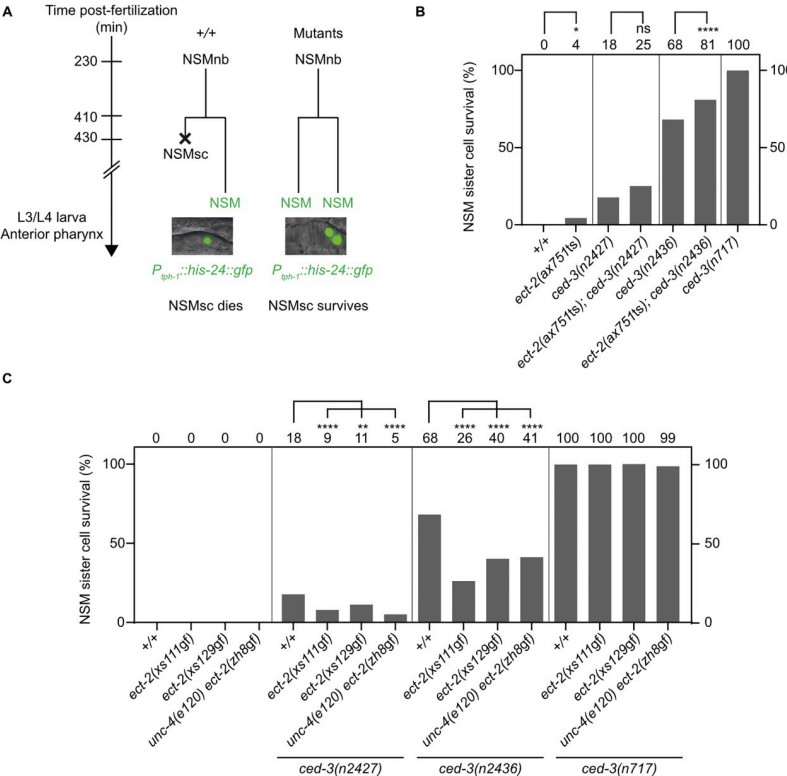

**Fig 5. *ect-2* RhoGEF promotes apoptosis in the NSM lineage. (A)** Schematic of NSM lineage and NSMsc survival assay. The differentiated NSM neuron can be identified in the anterior pharynx of L3/L4 larvae using the transgene *bcIs66* (P$_{tph-1}$::*his-24::gfp*). In wild-type (+/+), the NSMsc always dies, resulting in 1 NSM for each NSM neuroblast. In various mutants, the NSMsc survives, resulting in 2 "NSM"-like cells for each NSM neuroblast. **(B, C)** NSMsc survival (%) in various genotypes (*n* = 100–200). The NSMsc survival (%) is indicated on top of each bar graph. Statistical significance was determined using Fisher's exact test (**** = *P* < 0.0001, ** = *P* < 0.01, * = *P* < 0.05, ns = *P* > 0.05). NSM, neurosecretory motor neuron.

5C). However, we found that all 3 mutations suppress NSMsc survival caused by the partial *ced-3* lf mutations *n2427* or *n2436*, but not the null mutation *ced-3(n717)* (Fig 5C). For example, *ect-2*(*xs111*gf) significantly reduces NSMsc survival from 18% to 9% in *ced-3*(*n2427*) animals and from 68% to 26% in *ced-3*(*n2436*) animals. At least to our knowledge, this is the first reported suppression of the cell death abnormal (Ced) phenotype of animals homozygous for partial lf mutations of *ced-3* caspase. However, importantly, *ect-2*(*xs111*gf) does not act independently of *ced-3* caspase activity, because it fails to suppress the Ced phenotype of animals homozygous for the *ced-3* null mutation *n717*. In summary, decreasing *ect-2* RhoGEF function decreases the likelihood that the NSMsc undergoes apoptosis. Conversely, increasing *ect-2* RhoGEF function suppresses the NSMsc survival phenotype caused by partial lf mutations of *ced-3* caspase and therefore increases the likelihood that the NSMsc undergoes apoptosis. Together, these results demonstrate that in the NSMsc, *ect-2* RhoGEF has pro-apoptotic activity. The observation that increasing *ect-2* RhoGEF function does not suppress the NSMsc survival phenotype caused by a null mutation of *ced-3* caspase furthermore suggests that in the context of the apoptotic death of the NSMsc, *ect-2* RhoGEF acts upstream of and promotes *ced-3* caspase function.

To confirm that mutations of *ect-2* RhoGEF impact the death of other cells programmed to die through apoptosis, we analyzed the death of QL.pp using the reporter P$_{toe-2}$*gfp* (Fig 6A) [16,38]. The QL.p neuroblast divides unequally in L1 larvae to generate the smaller QL.pp and the larger QL.pa. QL.pp dies whereas QL.pa survives and divides to generate 2 cells that differentiate into neurons (PVM and SDQL) [9]. In L2/L3 larvae, P$_{toe-2}$*gfp* labels all QL.p descendants, which are 1 PVM neuron and 1 SDQL neuron in wild-type animals (Fig 6A). If QL.pp inappropriately survives, P$_{toe-2}$*gfp* labels either 1 PVM neuron, 1 SDQL neuron, and 1 "undead" QL.pp (PVM/SDQL neuron) or 2 PVM neurons and 2 SDQL neurons. P$_{toe-2}$*gfp* can therefore be used to determine % QL.pp survival (% animals with 3 or 4 rather than 2 P$_{toe-2}$*gfp*-positive cells). QL.pp survival in wild-type and *ced-3*(*n717*) animals is 0% and 97%, respectively (Fig 6B). Using *ect-2*(*ax751*ts), we found that the partial loss of *ect-2* RhoGEF causes 9% QL.pp survival. In addition, *ect-2*(*ax751*ts) increases QL.pp survival in animals homozygous for the weak *ced-3* lf mutation *n2427* from 39% to 73%. As predicted, animals homozygous for the 2 *ect-2* RhoGEF gf mutations *xs111* and *xs129* do not exhibit QL.pp survival (0%; Fig 6B). Importantly, each mutation can suppress QL.pp survival caused by the partial *ced-3* lf mutation *n2427*; however, neither suppresses QL.pp survival caused by the putative *ced-3* null mutation *n717* (Fig 6B). For example, *ect-2*(*xs111*gf) reduces QL.pp survival from 39% to 8% in *ced-3*(*n2427*) animals. However, *ect-2*(*xs111*gf) fails to suppress QL.pp survival in *ced-3*(*n717*) animals. These results indicate that *ect-2* RhoGEF has pro-apoptotic activity also in the QL.p lineage. As in the context of the apoptotic death of the NSMsc, *ect-2* RhoGEF acts upstream of and promotes *ced-3* caspase function in the context of the apoptotic death of QL.pp. Therefore, we conclude that the functional interactions between *ect-2* RhoGEF and *ced-3* caspase in the context of apoptosis are not restricted to the NSM lineage.

## Discussion

### Impact of apoptosis pathway on cell size

**CED-3 caspase de-recruits ECT-2 RhoGEF from the cell cortex on the dorsal side of the NSM neuroblast, thereby promoting unequal cell division.** The enrichment of ECT-2 RhoGEF on the plasma membrane on the anterior side of the 1-cell *C. elegans* embryo is the result of the de-recruitment of ECT-2 RhoGEF from the plasma membrane on the posterior side [39–41]. We previously showed that prior to NSMnb division, a gradient of CED-3 caspase activity can be detected in the NSMnb with higher CED-3 caspase activity on the dorsal side of

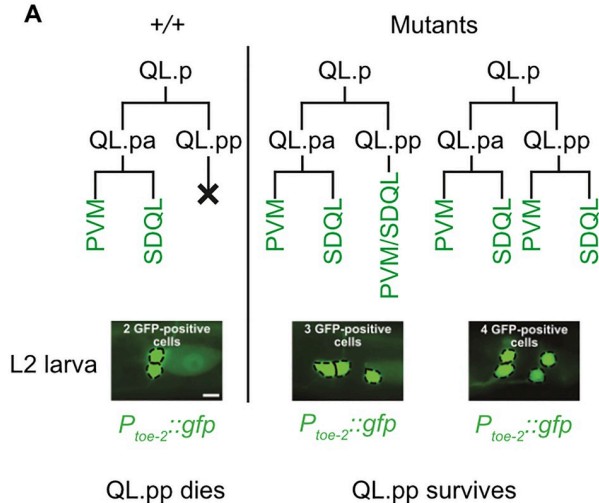

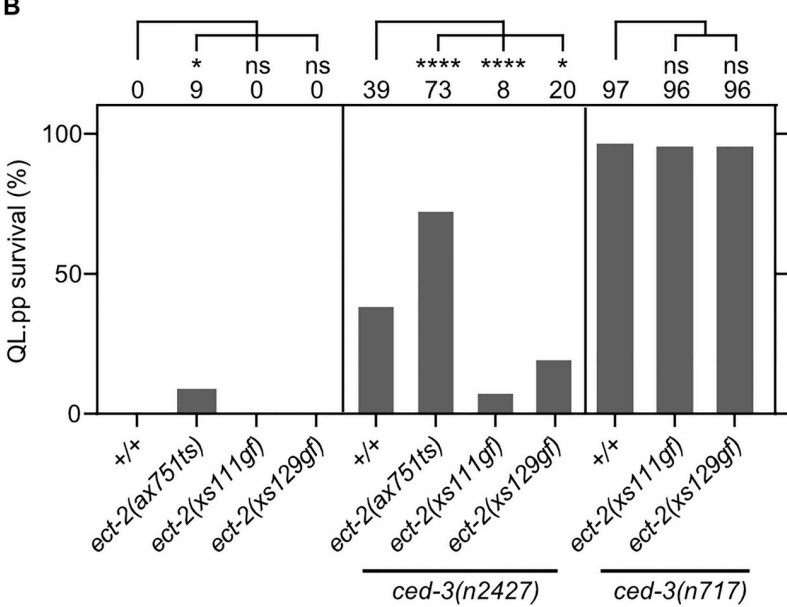

**Fig 6. *ect-2* RhoGEF promotes apoptosis in the QL.p lineage. (A)** Schematic of QL.pp survival assay. In wild-type (+/+), QL.pp dies and its sister cell QL.pa divides to form the neurons PVM and SDQL, which can be visualized in L2 larvae using the transgene *bcIs133* ($P_{toe-2}$::*gfp*). In various mutants, QL.pp survives and either differentiates into a PVM/SDQL-like neuron or divides to form 2 PVM/SDQL-like neurons. **(B)** QL.pp survival (%) in various genotypes ($n = 50–100$). The QL.pp survival (%) is indicated on top of each bar graph. Statistical significance was determined using Fisher's exact test. (**** = $P < 0.0001$, * = $P < 0.05$, ns = $P > 0.05$).

the NSMnb [15,42]. Importantly, the dorsal side is the side of the NSMnb where we observe less ECT-2 RhoGEF (Fig 4). Therefore, we propose that the cortical enrichment of ECT-2 RhoGEF on the ventral side of the NSMnb is the result of CED-3 caspase-dependent de-recruitment of ECT-2 RhoGEF from the plasma membrane on the dorsal side, rather than CED-3 caspase-dependent recruitment of ECT-2 RhoGEF to the plasma membrane on the ventral

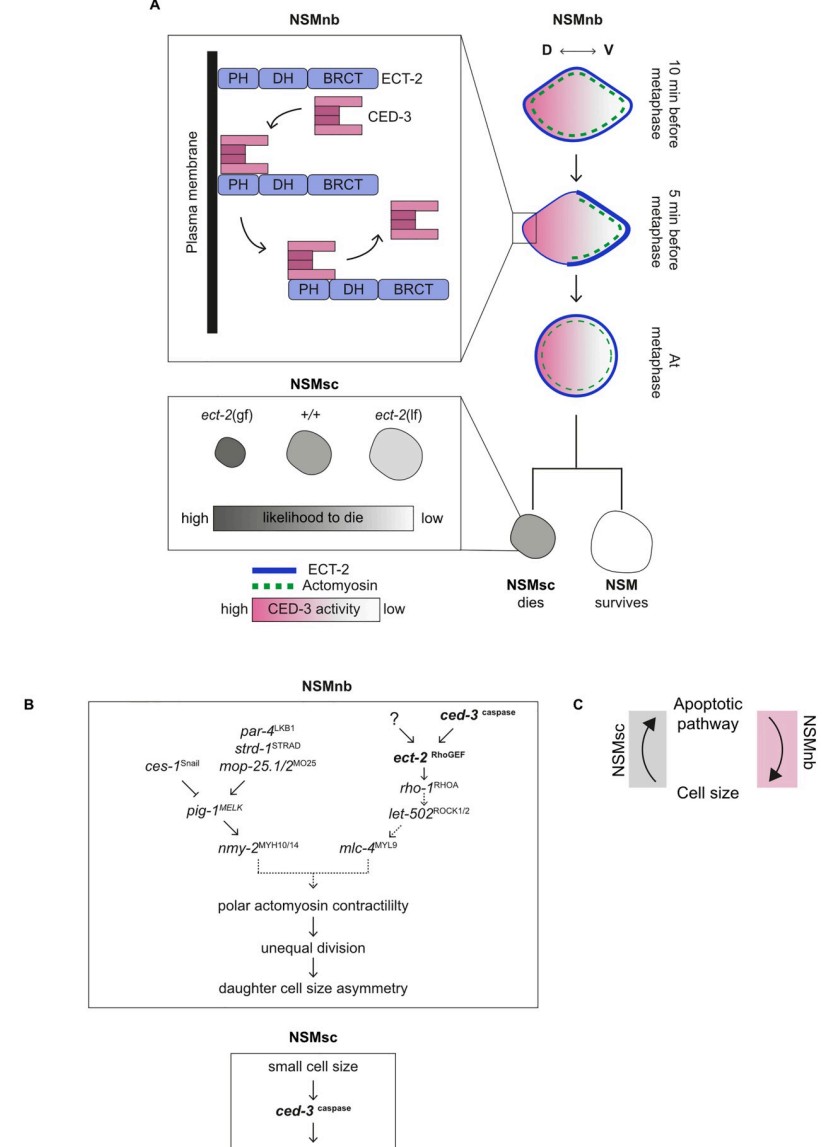

**Fig 7. Working model. (A)** Schematic representation of molecular and cellular events leading to unequal division of the NSMnb (top) and the subsequent death of the NSMsc (bottom). See text for further details. **(B)** Genetic pathways involved in sequential and reciprocal interactions between the apoptosis pathway and cell size in the NSM neuroblast (NSMnb) (top) and the NSM sister cell (NSMsc) (bottom). See text for further details. **(C)** Schematic indicating reciprocal interactions exist between the apoptotic pathway and cell size. See text for further details. NSM, neurosecretory motor neuron.

side (NSMnb; Fig 7A). Our in vitro binding experiments suggest that ECT-2 RhoGEF de-recruitment from the plasma membrane on the dorsal side is dependent on physical interactions between CED-3 caspase and ECT-2 RhoGEF and that these physical interactions are mediated by the PH domain of ECT-2 RhoGEF. Therefore, de-recruitment might be caused by impaired ability of the ECT-2 PH domain to bind phosphatidylinositol phosphates in the plasma membrane [43]. Although CED-3 caspase activity is necessary for de-recruitment, the reason for this requirement is uncertain, because ECT-2 RhoGEF is not readily cleaved by

CED-3 caspase in vitro. One possible explanation for our failure to detect CED-3 caspase-dependent cleavage of ECT-2 RhoGEF in vitro could be the absence in our in vitro binding experiments of a critical binding partner.

We previously provided evidence that a *pig-1* MELK, *nmy-2* nonmuscle myosin II-dependent pathway (which is controlled by both a *ces-1* SCRT-dependent and *par-4* LKB1, *strd-1* STRADα, *mop-25.1*, *.2* MO25α-dependent pathway) promotes polar actomyosin-dependent cortical contractility on the ventral side of the NSMnb prior to its division and that this is critical for unequal NSMnb division and daughter cell size asymmetry (NSMnb, Fig 7B) [14]. We now provide evidence that a *ced-3* caspase, *ect-2* RhoGEF-dependent pathway acts in parallel to the *pig-1* MELK, *nmy-2* nonmuscle myosin II-dependent pathway to ensure that the NSMnb divides unequally and that the NSMsc has a size below the critical lethal threshold (NSMnb, Fig 7B). Once activated (possibly through a *nop-1*, centralspindlin-dependent pathway [44,45]), ECT-2 RhoGEF on the ventral side of the NSMnb may therefore contribute to polar cortical contractility through the activation of the RhoA-like GTPase RHO-1, the "RhoA-associated coiled-coil containing protein kinase" (ROCK)-like kinase LET-502 (LET, lethal) and myosin light chain MLC-4 MYL9 (NSMnb; Fig 7B). However, the loss of *ced-3* caspase, which we found leads to the retention of ECT-2 RhoGEF on the dorsal side, does not affect the cortical enrichment of NMY-2 nonmuscle myosin II and F-actin on the ventral side [25]. This could indicate that ECT-2 RhoGEF may have targets other than RHO-1 RhoA in this context. Alternatively, the retention of ECT-2 RhoGEF on the dorsal side in *ced-3* lf mutants might not be sufficient to cause changes in the cortical enrichment of NMY-2 nonmuscle myosin II and F-actin; in support of this idea, the loss of *ced-3* caspase alone does not significantly affect the daughter cell size ratio in the NSMnb lineage (Fig 2D).

How is the activity of CED-3 caspase controlled in the context of the unequal cell divisions of mothers of cells programmed to die, such as the NSMnb? CED-3 caspase activity detected in mothers is dependent on the genes *egl-1* BH3-only and *ced-4* Apaf-1 and, hence, on the activation of the apoptosis pathway [15,16]. Furthermore, daughter cell size ratios in the NSMnb and QL.p lineages are affected not only by the loss of *ced-3* caspase but also by the loss of *egl-1* BH3-only and *ced-4* Apaf-1 [16]. This indicates that the entire apoptosis pathway contributes to the unequal cell divisions of mothers and the generation of smaller daughters with a size below the critical lethal threshold. The expression of *egl-1* BH3-only during development is finely tuned through a combination of transcriptional and post-transcriptional mechanisms, which results in low nonlethal levels of EGL-1 BH3-only and CED-3 caspase activity in mothers and high lethal levels of EGL-1 BH3-only and CED-3 caspase activity in their smaller daughter cells [17,46]. We propose that the role of the apoptosis pathway in unequal mother cell division is part of a developmental program that specifies the "cell death" fate during *C. elegans* development and that involves control of *egl-1* BH3-only expression.

## Caspases functionally interact with the actin cytoskeleton in both apoptotic and non-apoptotic contexts

Cells undergoing apoptosis exhibit cell shape changes, many of which are thought to be caused by caspase-dependent changes in the dynamics of the actin cytoskeleton [47]. For example, many types of apoptotic mammalian cells exhibit blebbing of the plasma membrane, and this is caused by localized contractility of the cortical actomyosin network. Blebbing is caused by caspase 3-dependent cleavage and activation of the kinase ROCK1, which phosphorylates and activates myosin light chain [48,49]. Caspase 3-dependent cleavage and activation of ROCK1 is also observed in non-apoptotic contexts such as in mammalian macrophages, whose cell shapes and, hence, functions are critically dependent on the actin cytoskeleton [50]. The actin

cytoskeleton is also a "non-apoptotic" target of the *D. melanogaster* caspase Dronc during the formation of actin-based cellular structures, such as sensory bristles and tracheal cell protrusions [51,52]. Furthermore, during Malpighian Tubule morphogenesis, the *D. melanogaster* caspase Drice has been reported to regulate Rho1 RhoA activity, thereby impacting critical cell shape changes and cell motility; both of these processes are dependent on dynamic changes in the actin cytoskeleton [53]. We provide evidence in support of the notion that in mothers of cells programmed to die during *C. elegans* development, *ced-3* caspase contributes to the polar activation of a pathway that most likely includes the *C. elegans* orthologues of RhoA (RHO-1) and ROCK1 (LET-502) and that promotes actomyosin contractility and unequal mother cell division. Thus, we have uncovered a new non-apoptotic role of *C. elegans ced-3* caspase. Furthermore, our work provides the first evidence of direct involvement of a caspase in unequal cell division, a developmental process that contributes to cell fate determination.

## Impact of cell size on activation of apoptosis pathway

**ECT-2 RhoGEF reduces the sizes of cells programmed to die, thereby enhancing CED-3 caspase activation and/or activity.** Mutations in several genes are known to affect the unequal divisions of mothers of cells programmed to die during *C. elegans* development. These mutations result in "unwanted" daughter cells that are larger in size and decrease the likelihood that such daughter cells undergo apoptosis. The affected genes have been implicated in the control of gene expression (*ham-1* STOX2, *ces-2* DBP TEF, *ces-1* SCRT, *dnj-11* DnaJ) [12,54,55], actomyosin contractility (*pig-1* MELK, *nmy-2* nonmuscle myosin II, *ect-2* RhoGEF) [11,14] (this study), receptor-mediated endocytosis (*grp-1* ArfGEF, *cnt-2* ArfGAP) [56,57], and cell signaling (*toe-2* DEPDC) [38]. We consider it highly likely that these mutations decrease the likelihood that an unwanted cell will undergo apoptosis by causing an increase in cell size. The following observation supports this notion. In the NSM lineage, the smaller daughter cell (NSMsc) is 0.66 times the size of the larger daughter cell (NSM) [12], and in the QL.p lineage, the smaller daughter cell (QL.pp) is 0.34 times the size of the larger daughter cell (QL.pa) [16]. In both lineages, the loss of *pig-1* MELK causes the mother cells to divide equally, resulting in daughter cells of essentially equal sizes [11,13]. Since daughter cell size ratios in the 2 lineages are different (0.66 compared to 0.34), the impact of the loss of *pig-1* MELK on the size of the smaller daughter cell is different in the 2 lineages: the loss of *pig-1* MELK leads to an approximately 1.25-fold increase in the size of the NSMsc but an almost 2-fold increase in the size of QL.pp. Interestingly, the loss of *pig-1* MELK causes a much weaker cell death defect in the NSM lineage compared to the QL.p lineage (2% NSMsc survival compared to 45% QL.pp survival). We consider it likely that this is the result of the relatively larger increase in size (in a *pig-1* lf background) of QL.pp. However, at this point we cannot rule out that mutations in *ect-2* RhoGEF (or any of the genes implicated in the unequal divisions of mothers of unwanted cells [see list above]) may decrease the likelihood that an unwanted cell will undergo apoptosis by causing defects in cell polarity or related processes rather than altering cell size.

Finally, with gf mutations of *ect-2* RhoGEF, we for the first time describe mutations that affect the unequal division of mothers and result in unwanted daughters that are smaller in size. The finding that these mutations increase the likelihood that these cells undergo apoptosis demonstrates that during *C. elegans* development, the size of an unwanted cell inversely correlates with its propensity to undergo apoptosis. Interestingly, this is consistent with the observation that within populations of animal cells grown in culture, smaller cells have a higher likelihood to undergo apoptosis [6]. Our finding also lends further support for the notion that cell size is a critical determinant of life versus death decisions during *C. elegans* development (NSMsc, Fig 7A).

Interestingly, there is increasing evidence that cell size is a critical determinant of life versus death decisions in the germline of adult *C. elegans* hermaphrodites as well, where more than 50% of the germ cells are eliminated through apoptosis [58]. The hermaphrodite germline is a syncytium, and germ cells share a common, central cytoplasm [59]. It was recently shown that changes in tissue hydraulics cause some germ cells to increase in size at the expense of other germ cells and that germ cells that decrease in size as a result subsequently undergo apoptosis [60]. Importantly, a decrease in the sizes of existing germ cells may also be caused through actomyosin-dependent contractility [61], which suggests that there may be unexplored parallels in the molecular mechanisms that control life versus death decisions in somatic cell lineages and the germline of adult hermaphrodites [1,58].

Through what mechanism(s) does a cell's volume affect the likelihood that it will die? We found that decreasing cell size suppresses the partial but not complete loss of *ced-3* caspase. Therefore, "small cell size" promotes *ced-3* caspase-dependent apoptotic cell death rather than a different type of cell elimination. In addition, this indicates that cell size exerts its influence either upstream of *ced-3* caspase or at the level of the enzymatic activity of its gene product. CED-3 caspase activity is known to be required and—once above a lethal threshold—sufficient for triggering apoptosis (NSMsc, Fig 7B) [29,62]. A decrease in cell size could lead to an increase in concentration of critical pro-apoptotic factors, such as EGL-1 BH3-only. It may also facilitate the assembly of CED-4 Apaf-1 dimers into functional apoptosomes and thereby promote proCED-3 caspase maturation and activation. Future work will be needed to address this important question.

A cell's size affects the likelihood that it will die; however, how cell size is controlled in this context and how cell size impacts a cell's commitment to the apoptotic fate have been unclear. Our work has uncovered novel, sequential, and reciprocal interactions between the apoptosis pathway and cell size in the context of programmed cell death during *C. elegans* development (Fig 7C). Specifically, we present evidence that by promoting unequal cell division, the *C. elegans* apoptosis pathway and its downstream effector *ced-3* caspase helps to generate "unwanted" cells with a size below a critical lethal size threshold. Conversely, by promoting CED-3 caspase activation or activity, "small cell size" promotes the elimination of unwanted cells through apoptosis. The non-apoptotic roles of caspases in the control of actin dynamics as well as the impact of cell size on the likelihood of a cell to undergo apoptosis appear to be conserved throughout the animal kingdom. For this reason, we speculate that the novel interactions between the apoptosis pathway and cell size uncovered in *C. elegans* may be conserved in higher organisms.

## Material and methods

### Strains and genetics

Unless noted otherwise, all *C. elegans* strains were cultured at 20˚C as described [63]. Bristol N2 was used as the wild-type strain. Mutations and transgenes used in this study are: LG I: *cp13(nmy-2::gfp+LoxP)* [64]. LGII: *ect-2(ax751*ts) [27], *ect-2(zh8*gf) [19], *ect-2(xs111*gf) [31], *ect-2(xs129*gf) [31], *unc-4(e120)* [63] and *zh135 (ect-2::gfp)* [32]. LG III: *strd-1(ok2283)* [35], *bcIs66* (P$_{tph-1}$*his-24::gfp*) [37]. LG IV: *ced-3(n2427)* [30], *ced-3(n2433)* [30], *ced-3(n2436)* [30], *ced-3(n717)* [29], and *pig-1(gm344)* [11]. LG V: *ltIs44* (P$_{pie-1}$*mCherry::ph$^{PLCδ}$*) [28]. Additional transgenes used in this study are: *bcIs133* (P$_{toe-2}$*gfp*) [16], *ddIs86* (P$_{pie-1}$*LifeAct::gfp*) [65].

### Plasmid construction

**pBC577** (S·TAG::CED-9): The full-length *ced-9* cDNA was cloned into the pCITE-4a(+) vector between the NcoI and EcoRI site [66]. **pBC1819** (proCED-3): The full-length *ced-3* cDNA was amplified from a cDNA library (Bristol N2) and cloned into the SmaI site of the

pBluescript II KS+ vector (this study). **pBC1820** (proCED-3(C358S): The G1073C point mutation was introduced into the *ced-3* cDNA through 1-site mutagenesis using plasmid pBC1819 as a template (this study). **pBC1922** (GST::proCED-3): The *ced-3* cDNA from pBC1819 was inserted into pGEX-4T1 via Gibson assembly [67] (this study). **pBC1923** (GST::proCED-3 (C358S)): The *ced-3*(G1073C) cDNA from pBC1820 was inserted into pGEX-4T1 via Gibson assembly (this study). **pBC1924** (GST::CED-3): The full-length *ced-3* cDNA was amplified from a cDNA library and cloned into the EcoRV site of pBluescript KS II(+) vector. It was then subcloned into the pCITE-4a(+) vector between the EcoRI and XmaI sites (this study). **pBC1926** (S·TAG::ECT-2): The *ect-2* cDNA was amplified from a cDNA library and cloned into the EcoRV site of pBluescript KS II(+) vector. It was then subcloned into pCITE-4a(+) vector between the NcoI and AvaI sites (this study). **pBC1964** (S·TAG::BRCT): The fragment of the *ect-2* cDNA encoding the BRCT domain of ECT-2 (339bp-879bp) was amplified from pBC1926 and cloned into the pCITE-4a(+) vector using Gibson assembly (this study). **pBC1965** (S·TAG::DH): The fragment of the *ect-2* cDNA encoding the DH domain of ECT-2 (1086bp-168 3bp) was amplified from pBC1926 and cloned into the pCITE-4a(+) vector using Gibson assembly (this study). **pBC1966** (S·TAG::PH): The fragment of the *ect-2* cDNA encoding the PH domain of ECT-2 (1710bp-2202bp) was amplified from pBC1926 and cloned into the pCITE-4a(+) vector using Gibson assembly (this study). **pET-CED-3** (proCED-3::FLAG) [68] was obtained as a gift from the lab of H.R. Horvitz, Massachusetts of Technology.

## Expression of recombinant proteins

Expression of recombinant proteins for the GST pull-down assay was performed as previously described [66]. BL21 (DE3) bacteria containing the GST-only pGEX-4T1 plasmid or the GST-tagged CED-3 plasmids (pBC1922, pBC1923, and pBC1924) were cultured at 37°C in LB medium containing 100 μg/ml carbenicillin until the $OD_{600}$ had reached 0.6. Protein expression was induced with 0.5 mM IPTG for 4 hours at 37°C, and 25 ml of cells were pelleted by centrifugation and the supernatant was discarded. Pellets were stored at −80°C for later use. Expression of recombinant proteins for the cleavage assay was performed as previously described [15,68]. BL21 (DE3) bacteria containing the pET-CED-3 or pET-3a plasmids were grown at 37°C in LB medium containing 100 μg/ml carbenicillin until the $OD_{600}$ had reached 0.6. Protein expression was induced with 1 mM IPTG for 2 hours at 25°C, and 50 ml of cells were pelleted by centrifugation and the supernatant was discarded. The pellets were stored at −80°C for later use. S-tagged CED-9, S-tagged ECT-2, and the various fragments of ECT-2 protein were synthesized in vitro in the presence of $^{35}$S-methionine using the TNT T7 Quick Coupled Transcription/Translation System (Promega) according to the manufacturer's instructions (Cat. No. L1170) (total volume of 50 μl).

## GST pull-down assay

GST pull-down experiments were performed as previously described [66,69]. The expressed protein pellets were resuspended in 2 ml of CED-3 extraction buffer (50 mM Tris-HCl (pH 8.0), 0.5 mM sucrose, and 5% glycerol, cocktail of protease inhibitors) and lysed by sonication (6 pulses of 10 seconds each with a 30-second interval at 20% amplitude using QSonica Q700) to release the proteins. The debris was collected by centrifugation and released GST-tagged proteins in the supernatant were purified using Glutathione-Sepharose beads (GE Healthcare) in the presence of CED-3 buffer for 90 minutes at 4°C and subsequently washed 3 times with CED-3 extraction buffer to remove unbound proteins. The purified GST-tagged proteins were incubated with 10 μl of in vitro transcribed-translated proteins in the presence of binding buffer (30 mM Tris-HCl (pH 7.5), 0.5% Triton X-100, 100 mM NaCl, 2 mM $MgCl_2$, 1 mM

DTT, 0.5% BSA, and a cocktail of protease inhibitors) for 90 minutes at 4˚C. The mixture was washed with binding buffer (without BSA) to remove unbound protein and the reaction was stopped by adding 2× LDS sample buffer (containing 5% β-mercaptoethanol) and heated to 70˚C for 10 minutes. All experiments were performed at least in triplicates.

### Cleavage assay

Cleavage assays were performed as previously described [15,68]. The expressed protein pellets were resuspended in 0.4 ml of CED-3 extraction buffer (50 mM Tris-HCl (pH 8.0), 0.5 mM sucrose, and 5% glycerol) and lysed by sonication (3 pulses of 5 seconds each with a 15-second interval at 20% amplitude using QSonica Q700) to release the proteins. The debris was collected by centrifugation and the resulting supernatant was used as lysate for the in vitro cleavage assays; 3 μl of CED-3 lysate was incubated with 3 μl of in vitro transcribed-translated proteins and 4 μl of CED-3 extraction buffer (total reaction volume of 10 μl) at 30˚C for 90 minutes. The pET-3a lysate was used as a negative control. The reaction was stopped by adding 10 μl of 2× LDS sample buffer (containing 5% β-mercaptoethanol) and heated to 70˚C for 10 minutes. All experiments were performed at least in triplicates.

### Analysis of GST pull-down and cleavage assay

Post-termination of the assay reactions using sample buffer, the reaction mixtures were run on a 10% Bis-TRIS gel (NuPAGE) using MOPS running buffer containing 0.1% SDS. The gels with the pull-down assay mixtures were stained with Coomassie staining solution followed by de-staining until distinct bands were visible. Gels were fixed using fixative (50% ethanol, 10% acetic acid), dried on a slab gel dryer at 80˚C, and exposed on either a photostimulable phosphor (PSP) plate or an X-ray film for detection of the $^{35}$S-labeled proteins. Exposed films were developed using a phosphor imager or X-ray film developer (PROTEC).

### Total mRNA extraction and Yeast-Two-Hybrid screening

To prepare a genomic cDNA library of Bristol N2 *C. elegans*, total mRNA was purified from populations of animals synchronized at different developmental stages (0 h embryo, 5 h embryo, 9 h embryo, L1 larvae, L2 larvae, L3 larvae, L4 larvae, mixed-stage adult) using the RNeasy Mini Kit (Qiagen) according to manufacturer's instructions (Cat. No. 74104). Next Interactions (https://nextinteractions.com/) generated the cDNA library using equal amounts of the mRNAs purified from each sample. The Yeast-Two-Hybrid screen was performed by Next Interactions (https://nextinteractions.com/).

### Confocal microscopy

Confocal imaging was performed as previously described using a Leica SP5 or SP8 microscope [14,15]. For red fluorescent proteins, the excitation wavelength was set at 561 nm, and the emitted light was collected between 567 to 662 nm using a HyD detector. For green fluorescent proteins, the excitation wavelength was set at 476 nm (*zh135*) or 488 nm (*cp13*, *ddIs86*), and the emitted light was collected between 500 to 550 nm using a PMT detector. The excitation laser power varied between different fluorescent proteins but was kept constant throughout the experiments. To mount embryos for imaging, 10 to 20 gravid adults were dissected in water to acquire mixed stage embryos. The embryos were transferred to 2% agarose pads, cover slips were placed on top and sealed with petroleum jelly. The embryos were incubated at 25˚C until the embryos reached the appropriate stage.

## Super-resolution microscopy

Super-resolution imaging was performed using a Zeiss LSM 980 with AiryScan2. For red fluorescent proteins, the excitation wavelength was set at 561 nm at 0.7%. For green fluorescent proteins, the excitation wavelength was set at 488 nm at 1.0%. The embryos were mounted, imaged, and analyzed similarly to the confocal microscopy protocol outlined above.

## Quantifying NSM sister cell survival

The percentage of surviving NSM sister cells was determined as previously described using the $P_{tph-1}his-24::gfp$ (*bcIs66*) transgene [37]. L3 or L4 larvae were mounted on 2% agarose pads containing 25 mM sodium azide in M9 buffer. The number of GFP-positive cells was counted in the anterior pharynx using a Leica Imager.M2 or Zeiss Axioscope 2. Wild-type worms contain 2 GFP-positive cells representing the 2 NSMs in the anterior pharynx and up to 2 extra GFP-positive cells can be seen in mutants (see Fig 3), representing inappropriately surviving NSM sister cells. The NSMsc survival percentage represents the number of NSM sister cells that inappropriately survived divided by the maximum number of NSM sister cells that could have survived.

## Quantifying QL.pp survival

The percentage of surviving QL.pp cells was determined as previously described using the $P_{toe-2}gfp$ (*bcIs133*) transgene, which specifically labels Q neuroblasts during the L1 larval stage [16]. Late L1 larvae (>5 hours post-QL.p division) were mounted on 2% agar pads using 10 mM levamisole in M9 buffer as paralytic agent. The number of GFP-positive cells was determined using a Zeiss Axioscope 2. Wild-type worms contain 2 GFP-positive cells representing the 2 QL.pa daughters (PVM and SDQL neurons). Up to 2 extra GFP-positive cells can be seen in mutants, representing inappropriately surviving QL.pp or QL.pp daughters. To validate the counting, all GFP-positive cells were observed in DIC to ensure they were not GFP-positive corpses. The QL.pp survival percentage represents the number of QL.pp that inappropriately survived (animals with 1 or 2 extra GFP-positive cells) divided by the sample size (number of animals analyzed).

## Determining daughter cell size ratio in the NSM lineage

The cell sizes of the NSMsc and NSM were calculated as previously described using the $P_{pie-1}mCherry::PH^{PLC\Delta}$ (*ltIs44*) transgene, which labels cell boundaries [12,15]. The cell size was estimated by summing up the area of each cell at different Z-slices (0.5μm step size) by drawing a region of interest (ROI) around the plasma membrane of the cell (see Fig 2A). The cell size of the NSMsc was divided by the size of the NSM to obtain "daughter cell size ratio." The images were obtained on a Leica TCS SP5 or SP8 microscope.

## Visualization and quantification of ECT-2::GFP, NMY-2::GFP, LifeAct::GFP, and mCherry::PH^PLCΔ in the NSM neuroblast

Visualization and quantification of the transgenes was performed as previously described [14]. The CRISPR knock-in allele *zh135 (ect-2::gfp)* [32] was used to visualize ECT-2. The CRISPR knock-in allele *cp13 (nmy-2::gfp+LoxP)* (Dickinson and colleagues, 2013) was used to visualize NMY-2. The transgene *ddIs86 (pie-1p::LifeAct::gfp, unc-119)* [65] was used to visualize F-actin. The transgene *ltIs44 (Ppie-1mCherry::PH^PLCΔ)* [28] was used to label cell boundaries to identify the NSM neuroblast. For all transgenes, quantification of fluorescence was performed using the central Z-slice of the NSM neuroblast. Using the *ltIs44* transgene, the central Z-slice of the

NSM neuroblast was divided into the dorsal and ventral halves by drawing a vertical line in the center of the neuroblast (S4 Fig) and an ROI for the 2 halves was saved on Fiji (ImageJ). Next, the mean gray value of the 2 saved ROI halves was measured in the GFP channel (*zh135* or *cp13* or *ddIs86*) or mCherry channel (*ltIs44*) on Fiji (ImageJ) (S4 Fig). The "Ventral/dorsal fluorescence ratio" was determined by dividing the ventral mean gray value by the dorsal mean gray value (also see S4 Fig). This was repeated for the 2 different time points, which were 5 minutes before metaphase ($t_{-5min}$) and at metaphase ($t_{0min}$). The metaphase time point ($t_{0min}$) was defined as the time point just before the NSM neuroblast undergoes cell division as the division was tracked live. Background subtraction was performed by drawing a $20 \times 20$ pixel ROI square in a random background area, measuring the mean gray value of this square, and subsequently subtracting this mean gray value from the central Z-slice used for quantification.

## Statistical analyses

Statistical analyses were performed using Prism by GraphPad. Normal distribution of the data was tested using the D'Agostino and Pearson test. If the data showed normal distribution, a 2 sample *t* test was performed to compare 2 independent groups (Welch's *t* test). For comparison of multiple groups showing normal distribution, a 1-way ANOVA was used with multiple comparison correction using Dunnett's T3 multiple comparison test. For the NSMsc and QL. pp survival percentages, statistical analyses were performed as a $2 \times 2$ contingency table using Fisher's exact test.

## Supporting information

**S1 Fig. Schematic representation of ECT-2 protein showing predicted CED-3$^{caspase}$ cleavage sites (vertical red line) and the amino acid sequence of the cleavage site in the PH domain, which is conserved across the *Caenorhabditis* species.**
(TIF)

**S2 Fig. (A) Schematic representation indicating the amino acid changes in the various *ect-2* alleles used in this study (B) Schematic representation of CED-3 protein (top) and *ced-3* gene (bottom) indicating the amino acid changes or nucleotide changes in the various *ced-3* alleles used in this study.**
(TIF)

**S3 Fig. Daughter cell size ratios in wild-type (+/+), *unc-4(e120)* and *unc-4(e120) ect-2(zh8)* mutants measured using the transgene *ltIs44*.** Each gray dot represents the daughter cell size ratio of 1 pair of daughter cells. The mean values are indicated using the horizontal red lines and are also provided on top. Statistical significance was determined using the Dunnett's T3 multiple comparisons test (**** = $P < 0.0001$, ns = $P > 0.05$).
(TIF)

**S4 Fig. Schematic representation of the methodology used to quantify ventral/dorsal fluorescence intensity ratio of the GFP fusion proteins shown in Fig 3 and S5 and S6 Figs.** The plasma membrane of the NSM neuroblast was marked using the transgene *ltIs44* ($P_{pie-1}$:: *mCherry*::PH$^{PLCΔ}$). The NSM neuroblast was divided into dorsal and ventral halves by drawing a vertical line along the center of the neuroblast. The mean fluorescence intensity of the appropriate transgene was measured in each half and divided to obtain their ratio. D is the dorsal side and V is the ventral side. Scale bar: 2 μm.
(TIF)

**S5 Fig. Ventral/dorsal ratios of mean GFP fluorescence intensities in the NSM neuroblast in animals carrying the CRISPR allele *zhIs135* (*ect-2::zf-1::gfp*) at 3 time points (t$_{-10min}$ = 10 minutes before metaphase, t$_{-5min}$ = 5 minutes before metaphase, t$_{0min}$ = metaphase).** Each gray dot represents the ventral/dorsal fluorescence intensity ratio of 1 NSM neuroblast. The mean values are given on top. Statistical significance was determined using the Welch's 2 sample *t* test (* = $P < 0.05$, ns = $P > 0.05$).
(TIF)

**S6 Fig. Ventral/dorsal ratios of mean mCherry fluorescence intensities in the NSM neuroblast in animals of indicated genotypes carrying the transgene $P_{pie-1}::mCherry::PH^{PLC\Delta}$ (*ltIs44*), which labels the plasma membrane of cells.** Each gray dot represents the ventral/dorsal fluorescence intensity ratio of 1 NSM neuroblast ($n = 12$). The mean values are indicated by the horizontal red lines and are also given on top. The horizontal black dotted line represents a fluorescence intensity ratio of 1, which indicates no asymmetry in fluorescence intensity between the ventral and dorsal side of the NSM neuroblast.
(TIF)

**S7 Fig. Daughter cell size ratios in wild-type (+/+) and various mutants as measured using the transgene $P_{pie-1}::mCherry::PH^{PLC\Delta}$ (*ltIs44*).** Each gray dot represents the daughter cell size ratio of 1 pair of daughter cells. The mean values are indicated using the horizontal red lines and are also provided on top. The horizontal red dotted line represents the mean daughter cell size ratio of wild-type (+/+) embryos for comparison. The horizontal black dotted line represents a daughter cell size ratio of 1.0 indicating equal division. Statistical significance was determined using the Welch's 2 sample *t* test (**** = $P < 0.0001$, *** = $P < 0.001$, ** = $P < 0.01$, * = $P < 0.05$, ns = $P > 0.05$).
(TIF)

## Acknowledgments

We thank members of the Conradt, Lambie, and Hajnal labs for discussions and comments on the manuscript. We thank M. Bauer, L. Jocham, N. Lebedeva, and L. McGuinness for excellent technical support; A. Hajnal and T. Kohlbrenner (University of Zurich, Switzerland) for allele *zh135*; and H.R. Horvitz (Massachusetts of Technology, USA) for plasmid pET-CED-3.

## Author Contributions

**Conceptualization:** Aditya Sethi, Hai Wei, Nikhil Mishra, Eric J. Lambie, Esther Zanin, Barbara Conradt.

**Data curation:** Aditya Sethi, Hai Wei, Nikhil Mishra, Ioannis Segos.

**Formal analysis:** Aditya Sethi, Hai Wei, Nikhil Mishra, Ioannis Segos, Barbara Conradt.

**Funding acquisition:** Barbara Conradt.

**Investigation:** Aditya Sethi, Hai Wei, Nikhil Mishra, Ioannis Segos, Barbara Conradt.

**Methodology:** Aditya Sethi, Hai Wei, Nikhil Mishra, Ioannis Segos, Eric J. Lambie, Esther Zanin.

**Project administration:** Barbara Conradt.

**Resources:** Hai Wei, Nikhil Mishra, Ioannis Segos, Eric J. Lambie, Esther Zanin, Barbara Conradt.

**Supervision:** Eric J. Lambie, Barbara Conradt.

**Validation:** Aditya Sethi, Ioannis Segos.

**Visualization:** Aditya Sethi, Hai Wei, Ioannis Segos.

**Writing – original draft:** Aditya Sethi, Barbara Conradt.

**Writing – review & editing:** Aditya Sethi, Hai Wei, Nikhil Mishra, Ioannis Segos, Eric J. Lambie, Esther Zanin, Barbara Conradt.

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
