## [Editor Report · Decision Letter 0]

10 May 2022

Dear Barbara, 

Thank you for submitting your manuscript entitled "Functional interactions between the apoptosis pathway and cell size are coordinated by the ced-3 caspase – ect-2 RhoGEF axis" for consideration as a Research Article by PLOS Biology.

Your manuscript has now been evaluated by the PLOS Biology editorial staff as well as by an academic editor with relevant expertise and I am writing to let you know that we would like to send your submission out for external peer review.

Once your full submission is complete, your paper will undergo a series of checks in preparation for peer review. Once your manuscript has passed the checks it will be sent out for review. To provide the metadata for your submission, please Login to Editorial Manager (https://www.editorialmanager.com/pbiology) within two working days, i.e. by May 12 2022 11:59PM.

If your manuscript has been previously reviewed at another journal, PLOS Biology is willing to work with those reviews in order to avoid re-starting the process. Submission of the previous reviews is entirely optional and our ability to use them effectively will depend on the willingness of the previous journal to confirm the content of the reports and share the reviewer identities. Please note that we reserve the right to invite additional reviewers if we consider that additional/independent reviewers are needed, although we aim to avoid this as far as possible. In our experience, working with previous reviews does save time. 

If you would like to send previous reviewer reports to us, please email me at ialvarez-garcia@plos.org to let me know, including the name of the previous journal and the manuscript ID the study was given, as well as attaching a point-by-point response to reviewers that details how you have or plan to address the reviewers' concerns. 

Kind regards,

Ines

--

Ines Alvarez-Garcia, PhD

Senior Editor

PLOS Biology

---

## [Decision Letter · Decision Letter 1]

27 Jun 2022

Dear Barbara,

Thank you for your patience while your manuscript entitled "Functional interactions between the apoptosis pathway and cell size are coordinated by the ced-3 caspase – ect-2 RhoGEF axis" went through peer-review at PLOS Biology. Your manuscript has now been evaluated by the PLOS Biology editors, an Academic Editor with relevant expertise, and by two independent reviewers.

The reviews are attached below. As you will see, the reviewers are very positive and find the conclusions of the manuscript interesting. Nevertheless, they both make several suggestions for improvement and to strengthen the results. After discussing the reviews with the Academic Editor, we are pleased to offer you the opportunity to address the comments from the reviewers in a revision that we anticipate should not take you very long. While we would like you to consider all the issues, we will leave up to you to address them either experimentally or textually.

We will then assess your revised manuscript and your response to the reviewers' comments with our Academic Editor aiming to avoid further rounds of peer-review, although might need to consult with the reviewers, depending on the nature of the revisions.

We expect to receive your revised manuscript within 1 month, but you can email us (plosbiology@plos.org) if you have any questions or concerns, or would like to request more time to perform some of the experiments. 

**IMPORTANT - SUBMITTING YOUR REVISION**

3. Resubmission Checklist

a) *PLOS Data Policy*

b) *Published Peer Review*

Best wishes,

Ines

--

Ines Alvarez-Garcia, PhD

Senior Editor

PLOS Biology

Reviewers' comments

Rev. 1: W. Brent Derry – Note that this reviewer has signed his review

The manuscript by Sethi et al. demonstrates a molecular connection between the C. elegans caspase CED-3 and the Rho guanine nucleotide exchange factor ECT-2 in establishing a cell size threshold for apoptotic death during embryonic development. Interestingly, this interaction does not result in the proteolytic processing of ECT-2. It is well established that somatic cells fated to die by apoptosis during C. elegans development are smaller than their surviving sister cells, but whether this is physiologically relevant has remained mysterious. Work from Dr. Conradt's lab previously showed mutants that cause symmetrical divisions of cells in lineages fated to die can suppress apoptosis, suggesting that a size threshold dictates life and death decisions. In this manuscript the authors present evidence that ECT-2 acts downstream of CED-3 to regulate asymmetric divisions, but upstream of CED-3 to promote apoptosis of the NSMsc and QL.pp neuroblasts. Overall, this is a solid manuscript and I only have a few suggestions/questions to help improve this excellent work.

1. The authors show that recombinant ECT-2 co-purifies with recombinant CED-3 (or proCED-3). Does this interaction dependent on whether ECT-2 is GDP or GTP bound? This is important to know because it would help determine if a signal that activates ECT-2 (to its GTP-bound form) is triggers a physical interaction with CED-3.

2. It is interesting that the Pleckstrin homology (PH) domain of ECT-2 binds CED-3. Normally, the PH domain interacts with phosphatidylinositol lipids. Have the authors checked if altering the phosphatidylinositol lipid content (for example, using an age-1 mutant) affects apoptosis and/or asymmetric divisions of the NSM or the QL lineage?

3. Given that CED-3 physically interacts with ECT-2 but does not cleave it, I wonder if this interaction affects the catalytic activity of CED-3? For example, does it stimulate or inhibit cleavage of CED-9?

4. Since cells normally fated to die by apoptosis often survive when they are a larger size (but not always) I am curious if the authors have considered that a pro-apoptotic factor such as EGL-1 or a microRNA specific for ced-9 is being diluted to prevent engagement of the apoptotic machinery? Can the authors estimate how the increased cell volume would affect the concentrations of these factors?

5. The authors suggest that ECT-2 might engage proteins other than RHOA/ROCK/NMY to regulate apoptosis of neuroblasts. Have they considered the RAS/MAPK pathway since ECT-2 has been shown to regulate this pathway during vulva development (PMID: 16270101)? Furthermore, the RAS/MAPK pathway also regulates developmental apoptosis (PMID: 11333233, 25144461). Although, it seems to have no role on NSMsc apoptosis (PMID: 25144461), it might have a role in the QL.pp lineage.

6. The authors propose that ect-2 acts downstream of ced-3 to promote asymmetric cell division, but upstream of ced-3 to promote apoptosis. This is a bit confusing as written. To make this point clearer I suggest they revise the model in Figure 7 to better reflect these activities.

7. The discussion is very long and should be shortened to be more concise.

Rev. 2:

In this manuscript from Sethi et al, the authors build on their previous work showing that in C elegans division of programmed to die mothers of cells die unevenly due to a short burst of polar cortical contractility of the actomyosin network directly before division. This is due to a very low, sub-lethal, level of caspase activity in these cells, and is dependent on BH3-only-, APAF- and caspase-like genes.

In this study, the authors convincingly show using Y2H and follow-up biochemical approaches that ECT-2 and ced-3 interact at the plasma membrane at the dorsal side in smaller cells, permitting an accumulation of ECT-2 on the ventral side, importantly uncovering a new, non-apoptotic role for ced-3. In general, the experiments are very well performed, well controlled and contain the appropriate statistics. Overall, this is a very nice manuscript.

However, for me, there is one question which is left unresolved, which if possible could be addressed further: why are larger cells less likely to undergo cell death? I assume this is due to the smaller effect sub-lethal caspase activity in these cells, but at least some more speculation in the discussion or some more experiments would be interesting to find out, such as addressing the question the authors themselves propose around APAF-1 dimerisation.

---

## [Decision Letter · Decision Letter 2]

8 Aug 2022

Dear Dr Conradt,

Thank you for the submission of your revised Research Article "Functional interactions between the apoptosis pathway and cell size are coordinated by the ced-3 caspase – ect-2 RhoGEF axis" for publication in PLOS Biology. I am writing on behalf of my colleague, Ines Alvarez-Garcia, who is away on vacation this week. On behalf of my colleagues and the Academic Editor, Ana J. Garcia-Saez, I am pleased to say that we can in principle accept your manuscript for publication. Your revised study was evaluated by the PLOS Biology editors, the Academic Editor, and by Reviewer 1, and we are all satisfied by the changes made. (Reviewer 1's comments are appended below my signature). 

While we are pleased to editorially accept your manuscript, before we can schedule it for publication we will need you to address any remaining formatting and reporting issues, which will be detailed in an email that you should receive within 2-3 business days from our colleagues in the journal operations team; no action is required from you until then. 

**IMPORTANT: As a last editorial request, after some discussion within the team, we think the title of your manuscript could be edited slightly for brevity and to make it more enticing to a broad readership. If you agree, we would propose that you change it to something like "A caspase–RhoGEF axis establishes the cell size threshold for apoptotic death in developing Caenorhabditis elegans". You can make this change to your manuscript as you address the other requests from our operations team.

PRESS

Sincerely, 

Lucas Smith, PhD

Associate Editor

PLOS Biology 

lsmith@plos.org

on behalf of

Ines Alvarez-Garcia, PhD

Senior Editor

PLOS Biology

REVIEWER COMMENTS:

Reviewer #1: I realize in hindsight that my first question was stupid because I was fixated on the activity of Rho, rather than the GEF ECT-2, so please ignore that one.

As for question 3, I agree that to prove binding of ECT-2 to CED-3 affects its cleavage activity would be very difficult and is beyond the scope of this paper. In my mind I was wondering if simply adding purified ECT-2 back to the mixture would affect the cleavage of CED-9 (as shown in Fig. 1E), but there are many potentially complicating problems when using lysates and semi-purified proteins. 

Overall, I am satisfied with the explanations provided by the authors and think the manuscript is acceptable for publication in its present form.